# Pipeline for the Generation and Characterization of Transgenic Human Pluripotent Stem Cells Using the CRISPR/Cas9 Technology

**DOI:** 10.3390/cells9051312

**Published:** 2020-05-25

**Authors:** Joffrey Mianné, Chloé Bourguignon, Chloé Nguyen Van, Mathieu Fieldès, Amel Nasri, Said Assou, John De Vos

**Affiliations:** 1IRMB, Univ Montpellier, INSERM, CHU Montpellier, 34000 Montpellier, France; joffrey.mianne@inserm.fr (J.M.); chloe.bourguignon@inserm.fr (C.B.); nguyenva@etud.insa-toulouse.fr (C.N.V.); m_fieldes@hotmail.com (M.F.); amel.nasri@inserm.fr (A.N.); 2Department of Cell and Tissue Engineering, Univ Montpellier, CHU Montpellier, 34000 Montpellier, France

**Keywords:** hPSC, CRISPR, screening, stem cells, gene editing, gene targeting, disease modeling

## Abstract

Recent advances in genome engineering based on the CRISPR/Cas9 technology have revolutionized our ability to manipulate genomic DNA. Its use in human pluripotent stem cells (hPSCs) has allowed a wide range of mutant cell lines to be obtained at an unprecedented rate. The combination of these two groundbreaking technologies has tremendous potential, from disease modeling to stem cell-based therapies. However, the generation, screening and molecular characterization of these cell lines remain a cumbersome and multi-step endeavor. Here, we propose a pipeline of strategies to efficiently generate, sub-clone, and characterize CRISPR/Cas9-edited hPSC lines in the function of the introduced mutation (indels, point mutations, insertion of large constructs, deletions).

## 1. Introduction

Thanks to their unlimited proliferative potential, their euploid state, and their capacity to differentiate in any cell type, human pluripotent stem cells (hPSCs), either embryonic or induced, have a tremendous potential for disease modeling and for the production of cells for clinical applications [1,2,3]. Although numerous hPSC lines from patients with various diseases have been established, the generation of hPSC lines for some pathologies or rare genetic mutations is still challenging. Moreover, interindividual genetic heterogeneity can result in biological variations that may render the comparison between lines difficult, especially between hPSCs derived from healthy controls and patients [4,5]. The ability to genetically manipulate hPSCs offers the opportunity to introduce, modify, or correct mutations and to generate genetically matched isogenic control lines, to establish definitive genotype–phenotype links [6,7].

Recent technologies based on site-specific nucleases, including zinc finger nucleases (ZFNs), transcription activator-like effector nucleases (TALENs), and particularly the clustered regularly interspaced short palindromic repeats (CRISPR) system, have enabled the versatile genome engineering of hPSCs [8,9]. Nevertheless, hPSC engineering remains challenging due to the intrinsic properties of such cells, including their relatively poor transfection efficacy and low viability following transfection, difficulty in isolating clonal populations, preferential selection and amplification of aneuploid clones, and spontaneous cell differentiation. To alleviate these problems, several protocols to generate an array of different mutagenesis events have been described [10,11,12,13,14]. While much effort has been focused on improving the methodological procedures to generate transgenic hPSCs, only few studies assessed the key quality control steps, including screening and in-depth molecular characterization. Yet, appropriate validation is fundamental to establish the relevance of the edited cell lines, but it is made difficult by the high heterogeneity of the allele types that can be generated by CRISPR-based engineering [15,16]. Here, we summarize the main strategies and propose an efficient pipeline to generate, screen, and characterize genetically engineered hPSC lines obtained by knock-out (KO), small or large knock-in (KI), and deletion of genetic elements.

### 1.1. The CRISPR/Cas9 System

The discovery and adaptation of natural biological systems as site-specific nucleases have revolutionized the field of genetic engineering in the past two decades. The implementation of meganucleases, ZFN, TALEN, and now CRISPR/Cas systems has expanded the opportunities to manipulate the genome of many species, including human cells [17]. Particularly, the CRISPR/Cas system has become the main tool for genome engineering projects due to its simplicity, versatility, efficacy, and low cost [17].

The CRISPR/Cas9 system, derived from *Streptococcus pyogenes* (spCas9), is the first CRISPR system to have been adapted for gene editing [18]. It relies on a universal Cas9 nuclease that can generate a DNA double-strand break (DSB) when combined with a single-guide RNA (sgRNA) to form a ribonucleoprotein (RNP) complex [18]. In this RNP complex, the sgRNA will guide the Cas9 nuclease to a specific locus by Watson–Crick base pairing, thus allowing nuclease activity and cleavage of the target site (Figure 1). The sgRNA can be designed to target any 20-nucleotide-long sequence that must be followed in the targeted genome by a 5′-NGG tri-nucleotide recognition site, called a protospacer adjacent motif (PAM) [18].

Although many CRISPR/Cas systems derived from various bacteria or engineered to recognize broader sets of PAMs, to be more efficient or more specific, have now been adapted as site-specific nucleases, this review will only focus and describe the most commonly used spCas9 [19]. However, the strategies and recommendations proposed in this review are applicable to the different CRISPR systems adapted from spCas9 or other DNA-targeting Cas proteins.

### 1.2. DNA Repair Mechanisms

Introducing a DNA DSB at a targeted locus will lead to activation of the cell endogenous DNA repair mechanisms. Three mechanisms are predominantly activated (Figure 1) [20]. The non-homologous end-joining (NHEJ) and micro-homology-mediated end-joining (MMEJ) pathways are usually considered as error-prone systems. Therefore, by taking advantage of these DNA repair mechanisms, it is possible to introduce small insertion or deletion (indel) events that lead to the disruption of the targeted DNA sequence. On the other hand, the homology-directed repair (HDR) pathway can be exploited to introduce precise nucleotide modifications or exogenous DNA sequences by providing a DNA donor template with homology to the target site. One of the drawbacks of relying on cell endogenous DNA repair mechanisms is our limited ability to preferentially select one of them [20]. This is particularly limiting because HDR events tend to occur at a much lower rate than NHEJ-MMEJ events.

## 2. Preparing the Experiment

For successful CRISPR-based mutagenesis of hPSCs, it is important to carefully design and plan the experiment. Specifically, before starting laboratory experiments, the following points should be assessed.

### 2.1. Defining the Project Goal

Clearly defining the project goal is essential for selecting the most time- and cost-efficient approach to obtain the desired cell line. This means specifying the type and purpose of the cell line(s) to be generated. Particularly, it is important to know whether a clonal cell line is required, whether the targeted gene is expressed in and/or is essential for hPSC maintenance, or whether it is expressed only upon hPSC differentiation, and whether the obtained cell lines will be used for basic research, pre-clinical, or clinical purposes. Answering these key questions will ensure the project feasibility, will guide reagent selection, and help to define the quality control (QC) level required to validate the cell line for downstream experiments.

### 2.2. Defining the Mutagenesis Event to Be Generated

Various type of mutants can be generated using CRISPR technologies. Genes can be KO, single nucleotide polymorphisms (SNPs) can be introduced or corrected, large constructs can be KI to add functional elements (e.g., constitutive protein expression, fluorescent reporters, tags, conditional alleles, inducible systems, etc.), or particular sequences can be deleted. Several mutagenesis events can be obtained in a single experiment. These events can be obtained separately as heterozygous or homozygous variants but can also be found in combination (compound-heterozygosity, e.g., one KO and one KI allele). Determining the mutagenesis event(s) to be generated will specify the experiment design and the screening process to recover the cells with the desired mutation(s).

### 2.3. Selecting Reagents and Transfection Strategies

The selection of the relevant reagents and transfection strategies are important because this will directly influence the experiment design and the number of QC tests to be performed. The sgRNA/Cas9 reagents can be delivered under various forms, such as plasmids, RNA, or RNP. Plasmids are stable constructs that can be easily engineered to co-express multiple sgRNAs at once, or additional functions (e.g., fluorescent markers or antibiotic resistance). The addition of fluorescent reporters or antibiotic resistance cassettes can help to assess the transfection efficacy and allow the positive selection of transfected cells. Expression plasmids should carry suitable promoters to avoid transcriptional silencing that can occur in hPSCs [21]. Moreover, when using plasmid expression systems, DNA toxicity, potential unwanted insertion into the genome, and prolonged expression that can increase the chance of introducing undesired CRISPR-induced off-target (OT) mutations should be taken into account [22]. Alternatively, the sgRNA/Cas9 complex can be delivered as RNA or RNP. Although they usually do not allow the co-expression of fluorescence markers or antibiotic selection, they alleviate most of the disadvantages inherent to plasmid transfection due to their low toxicity, absence of genomic integration, and quick transient expression [22]. If required, a DNA donor template can also be added to the system, such as circular or linear double-stranded DNA (dsDNA), and linear single-stranded DNA (ssDNA). dsDNA are easy to synthetize, to manipulate, and can be used to deliver large constructs but come with higher risk of unintended integration into the targeted genome (backbone integration at the targeted locus or random insertion) [16]. Both short (up to 200nt) and long (>200nt) ssDNA are highly efficient templates for homologous recombination, and their single strand nature reduces the odds of unintended integration [14,23,24,25].

These components can be introduced in cells by electroporation [10], lipofection [26], using biopolymer systems [27], or by viral transduction [28]. The advantages and limitations of each strategy have been discussed elsewhere [29,30]. It is worth noting that electroporation and lipofection are the main transfection approaches for CRISPR/Cas9 gene-editing in hPSCs. Other systems can be interesting but might require optimization for hPSC transfection.

### 2.4. Characterizing the Targeted Sequence

Although important, the characterization of the hPSC genomic sequence to be modified is often neglected, leading to a waste of time and resources. Indeed, due to the inter-individual genetic heterogeneity, it is not rare to observe small or large nucleotide variations (SNP, insertions, deletions) in one or both alleles compared with the reference genome [31]. Analysis of both alleles by Sanger sequencing before sgRNA selection will ensure that the sgRNA(s) and DNA donor templates are designed in accordance with the actual target sequence.

### 2.5. Anticipating the Screening Strategy

Another important point in the experiment design is the downstream screening strategy. Screening should be inexpensive and high throughput to allow the processing of as many clones as required (depending on the mutation event complexity and the overall efficacy of the CRISPR/Cas9 system). This is especially true for KI projects that depend on HDR, which is less effective than NHEJ-MMEJ. Strategies to facilitate screening or to reduce the number of clones to be screened should be considered during the experiment design. The selected clones will then require further characterization using more specific but more expensive and time-consuming tests.

### 2.6. sgRNA Design and Selection

sgRNA selection is a multifactorial evaluation process that can be described as a compromise between specificity and efficacy/probability to obtain the desired modification. For spCas9, sgRNA should recognize a target sequence of 18-20 nucleotides that must be followed by the trinucleotide 5′-NGG PAM sequence at the target site [18]. PAM sequences can differ depending on the chosen CRISPR system [19]. However, irrespective of the CRISPR system to be used, the selection parameters are specific for the modification to be generated. Thus, for protein loss-of-function projects, it is recommended to select sgRNAs that (i) will disrupt a sequence present in all splice variants; (ii) are not too close to the start codon to avoid activation of a downstream alternative start codon; (iii) are not too close to splice-sites, which if deleted, can result in unpredictable new transcripts; and (iv) are as specific as possible, preferentially without OT sites with three or less mismatches. For KI projects, sgRNA selection is more constrained because the DSB should occur as close as possible to the site of the nucleotide change (typically within 10 nucleotides of either side of the DBS) [14,23,25]. The sgRNA should also be selected to avoid DNA donor cleavage and reprocessing of the allele to be generated following the homologous recombination event. Like for KO projects, the selected sgRNA must be as specific as possible. For deletion projects, the selection of a pair of sgRNAs depends on the nature of the deletion to be generated. If a precise sequence must be deleted (with a defined number of nucleotides to be removed and a precise junction point), then a donor DNA template might be used, and sgRNAs should be selected like described for KI projects [32]. If the deletion does not need to result in a precise junction event, sgRNAs should be selected to introduce a DSB in the region surrounding the sequence to be deleted, and should be as specific as possible.

Many online tools using various algorithms have been developed and are available to help with the experiment design and sgRNA selection depending on their specificity [33]. Moreover, several algorithms can be used to predict sgRNA efficacy and the type of indels that are generated following NHEJ-MMEJ repair of DBS [33,34]. Although these algorithms can be useful to prioritize sgRNA selection, they only give predictions and still have many limitations. Therefore, as sgRNA efficacy can be highly variable, it is important to design two-three sgRNAs and in vitro screen their efficacy by T7 endonuclease 1 (T7E1) assay or by tracking of indels by decomposition (TIDE) analysis with the aim of identifying the most efficient [35,36].

### 2.7. DNA Donor Template Design

For projects in which exogenous nucleotides will be KI, a DNA donor template must be generated. Its design depends on all of the factors described above, and particularly on the modification to be introduced and the selected sgRNA. The DNA donor template must include the precise sequence to be introduced and also the homology arms to the target. It may also include additional sequences to facilitate the recovery of correctly edited clones. The recommended length for the homology arms is 20-50nt for small ssDNA [14,23], 100-600nt for long ssDNA [24,25], and 300-1000nt for dsDNA (PCR products or plasmids) [13,37]. For ssDNA, phosphorothioate modifications at both donor sequence ends are recommended to improve the ssDNA stability and KI efficacy [23]. The DNA donor template should not include the sgRNA sequence to be used. This can be obtained by selecting sgRNA sequences that will be disrupted by the nucleotides to be introduced, or by adding silent mutations that will prevent tight binding of the sgRNA and facilitate the downstream screening by introducing an enzymatic restriction site.

Of note, strategies based on homology-independent targeted integration have also been described [38]. This might be an interesting alternative; however, their efficacy in hPSC remains to be determined because only limited reports are available [39].

## 3. Experimental Strategies for the Successful Generation and Characterization of CRISPR-Edited hPSC Lines

Irrespective of the desired modification, the generation of transgenic hPSC lines can be decomposed in four main steps: Transfection, clone isolation, screening, and in-depth molecular validation (Figure 1). Each step can be achieved using various strategies that will be discussed below.

### 3.1. Transfection

The first step consists in delivering the reagents (sgRNA, Cas9, and DNA donor) into the cells. Its success is influenced by many factors, including the choice of transfection method and the cell line susceptibility. Various tools and protocols have been specifically adapted for the efficient transfection or transduction of hPSCs [26,28,40]. Transfection approaches using electroporation systems or lipofectamine are cheap, have been widely used and optimized, and allow the simultaneous transfer of all reagents (sgRNA, Cas9, and DNA donor) in all possible forms (DNA, RNA, RNP). Other transfection strategies based on polymers or nanoparticles have also been used to deliver CRISPR reagents. However, these approaches have rarely been tested or optimized for hPSC transfection. Transduction approaches using viral vectors are interesting for their high efficiency, but they usually remain more expensive [28]. Moreover, the inability to package the CRISPR reagents and the donor template into the same particle remains a limitation of transduction strategies [28].

Several points should be taken into account to improve the transfection efficacy. First, hPSCs should be “adapted” to single-cell passaging using the Rho-associated coiled-coil containing kinases (ROCK) inhibitor Y-27632 for four to six passages before starting the experiment [41]. This will ensure optimal cell survival after transfection, and will optimize clone selection. The number of cells to be transfected depends on the transfection system toxicity and the likelihood of obtaining the desired event (depending on its complexity and sgRNA efficacy). Toxicity is also influenced by the quality and quantity of nucleic acids and/or proteins to be transfected. Optimizing these parameters will significantly increase the probability of obtaining the desired event(s). Additionally, it has been reported that hPSC mortality via activation of the tumor protein p53 apoptosis pathway is high after DNA DSB introduction in their genome [42]. It is therefore important to scale up appropriately the number of cells to be transfected. Finally, to promote cell survival after transfection, it is also recommended to culture cells in medium supplemented with the ROCK inhibitor Y-27632 for 24 to 48h [41].

### 3.2. Clone Isolation

CRISPR-based mutagenesis of a pool of cells will lead to a variety of mutagenesis events. Each cell within the pool will harbor its own combination of wild-type and/or mutant alleles. Although heterogeneous cell populations can be used for some research projects, in most cases, clones that harbor only the desired mutagenesis event must be isolated. This can be obtained by seeding single cells in the wells of 96-well plates. Single cells can be isolated manually by limiting the dilution approach, or by automated cell sorting. Alternatively, a pool of single cells can be plated at low density and later, colonies can be picked and manually transferred in individual wells.

For some projects, clone isolation can be facilitated by adding a positive selection marker (fluorescent reporter, or antibiotic resistance gene) to specifically select the transfected, or correctly engineered cells. Cells that express a fluorescent reporter can be selected by fluorescence-activated cell sorting (FACS), whereas cells that include an antibiotic resistance gene will survive antibiotic selection. These positive markers are usually transiently expressed as part of the transgene and removed at a later stage, or are delivered with the transient Cas9 expression system. If the project aim is to introduce a tagged protein or a gene, the expression of which is controlled by an endogenous promoter, the strategy can be adapted depending on whether this gene is expressed or not in hPSCs. Indeed, if the protein with a fluorescent tag is expressed in hPSCs, correctly targeted cells can be detected and isolated by FACS [13]. Conversely, FACS-based screening is not possible if the tagged protein is not expressed in hPSCs. Innovative approaches have been proposed to overcome this issue. For example, Roberts et al. described a strategy based on a transgene with a constitutively expressed mCherry fluorescence selection cassette that allows selection by FACS of clones that carry the transgene. This was followed by excision of the selection cassette using CRISPR/Cas9 and MMEJ repair [12].

Even when adapted to single-cell passaging, hPSCs remain highly sensitive to cell dissociation and isolation. Consequently, when placed in individual wells, most cells will die within few days. Plating cells at low density will improve survival but will also increase the probability of forming mosaic colonies, derived from two or more genetically different cells. Therefore, irrespective of the used approach, the number of clones to be isolated should be scaled up according to the mortality but also to the likelihood of obtaining the desired event.

Following clone isolation, an intermediate step to expand and duplicate clones is required to maintain cells in culture or to cryopreserve them, while the clone DNA is extracted for screening.

### 3.3. Clone Screening

Screening can be defined as a single or a combination of high-throughput tests to rapidly and cheaply identify the clones of interest. Two main parameters should be considered when selecting a screening test: (i) The number of clones to be screened, and (ii) the type of modifications introduced (Figure 2 and Figure 3).

When indels are introduced, different strategies can be employed for clone screening, each of them with its own advantages and limitations (Table 1).

T7E1 or Surveyor nuclease assays are based on the PCR amplification of the targeted sequence, followed by a melting step and random hybridization that can form homo- or hetero-duplexes if one or both alleles within that clone are mutated (the two alleles need to be different for detection). For homozygous clone detection, DNA from the parental cell line must also be amplified and supplemented to the PCR product. Then, these homo- or hetero-duplexes undergo T7E1 or surveyor nuclease enzymatic digestion that will specifically cleave heteroduplexes, thus revealing the mutated clones after electrophoresis [35]. The main limitation of these tests is their low sensitivity to detect some mutations, including single nucleotide variations that are one of the main indel types generated by NHEJ-MMEJ [34,35]. High-resolution melting analysis (HRMA) can also be used for clone screening [43]. This approach consists in following in real time the fluorescence emission of an intercalating dye during the melting of a short PCR product that includes the targeted sequence. Mutated clones will be detected due to the slight modification of their melting curve (Figure 2) [43]. This test is inexpensive and convenient to quickly identify mutants in a large set of clones. Other approaches, such as indel detection by amplicon analysis (IDAA) and droplet digital PCR (ddPCR), can be used for indel identification [44,45]. Both tests are more expensive because they require fluorescent primers or probes. However, IDAA present the advantage of giving information on the allele composition by providing the number of nucleotides added or deleted compared with the wild-type allele (of note, it only allows the detection of size differences but not nucleotide composition changes) [44]. This is particularly interesting for KO projects to select clones in which both alleles are mutated while avoiding in-frame deletions.

For deletion projects, screening usually relies on PCR amplification using primers that surround the region to be deleted to detect the corresponding shorter product after electrophoresis (Figure 2). However, when the deletion size is large, alleles without a deletion event might be difficult to amplify by PCR. Therefore, it is important to combine PCR screening with a copy counting assay by quantitative PCR (qPCR) or ddPCR, to confirm zygosity.

For small KI projects (e.g., introduction or correction of an SNP), the screening strategy depends on the donor design. Introducing a restriction site within the donor facilitates the screening of clones that have integrated the donor on the target. Indeed, positive clones can be identified by PCR amplification of the targeted region followed by enzymatic digestion and electrophoresis (Figure 2). If no restriction site can be introduced in the donor, then primers that will only amplify the expected allele can be designed.

For large KI projects, screening is performed by combining different tests (Figure 3). This includes PCR amplifications using primers that bridge between the transgene and the surrounding genomic DNA (bridge_PCR) to detect clones with on-target transgene integration. An additional PCR to amplify alleles that have not integrated the transgene (NT_PCR) is also performed. Large transgenes are usually delivered as plasmids that may integrate randomly in the genome or in concatemers at the target site [13,16]. To exclude such events, the transgene copy (TC) number must be evaluated by qPCR or ddPCR. Together, the PCR data and TC counting will discriminate clones into four categories: No transgene integration (bridge_PCR negative, NT_PCR positive, TC = 0), heterozygous integration (bridge_PCR positive, NT_PCR positive, TC = 1), homozygous integration (bridge_PCR positive, NT_PCR negative, TC = 2), and others (all the other possibilities, including random transgene insertion and mosaic clones). Alternatively, Southern blot analysis can be performed to identify clones with on-target transgene integration after bridge_PCR.

For all mutation types (KO, deletion, small or large KI), positive clones recovered after these first screening steps will need to be further analyzed by Sanger sequencing to determine the exact allelic sequence composition (Figure 2 and Figure 3). This will allow the selection of only clones with the correct on-target event for downstream in-depth molecular characterization. For homozygous clones, a single sequencing trace will be obtained, directly revealing the sequence of both alleles. Nevertheless, the ploidy status of homozygous clones should be confirmed by qPCR or ddPCR because large deletion events may sometimes result in hemizygosity (Figure 2) [46]. For heterozygous or compound-heterozygous clones, two overlapping sequences will be obtained and might require sub-cloning to determine the exact sequence of both alleles (Figure 2).

For gene KO projects, only clones in which both alleles contain the frameshift mutation will be kept for further analysis. For clones harboring a deletion, sequencing will give information on the exact sequence that has been deleted, and on the absence of other unwanted modifications (small insertion, inversion) that may be generated at the break points. For KI projects, sequencing is important to confirm the integrity of the newly generated allele, because unperfect repair using donor templates can occur [23,24,25,47]. Moreover, for heterozygous clones in KI projects, the allele without donor sequence integration should also be characterized because indels may be present [13].

Of note, if only a few clones have to be analyzed, it is possible to directly check them by Sanger sequencing, thus bypassing some of the screening steps.

By the end of the screening process, the clones of interest should have been selected and cryopreserved. Among them, two to five candidates will be selected for in-depth molecular characterization.

### 3.4. In-Depth Molecular Characterization

At this stage, additional analyses of the selected clones are required to fully characterize and validate them (Table 2). The type and number of tests will depend on the reagents used and the quality level required for the downstream applications. These quality control steps are important to ensure that no unintended event, which may interfere with the genotype–phenotype correlation analysis, has occurred during genome engineering and clonal selection.

Unintended events associated with the mutagenesis process include random genomic integration of the donor DNA or of the used expression systems [13,16,48]. If plasmids have been used to express the CRISPR/Cas9 system or as a donor DNA template, PCR analysis with primers targeting the plasmid backbone should be performed to determine whether they have been integrated. Similarly, the random integration in the genome of the short and long ssDNA used as donor templates should be assessed by copy counting using qPCR or ddPCR [24,25].

Another issue that can arise during CRISPR/Cas9 mutagenesis is the possible generation of OT mutations due to the imperfect specificity of the system [49]. This can occur when the used sgRNAs have a low specificity profile. OT mutations are also more likely to be generated if Cas9 activity is prolonged [22]. Therefore, the absence of OT mutations should be confirmed, especially if plasmid expression systems have been used because of prolonged exposure to Cas9 activity. Many approaches and protocols have been developed to this aim [50]. While most projects will only require the checking of a few sites that are most likely to be modified by HRMA or Sanger sequencing (top 10 OT sites, or all OT sites with up to three mismatches associated with the used sgRNA), others might need a genome-wide analysis of all possible OT sites [50] or next-generation sequencing approaches (whole-exome or whole-genome sequencing). Another possibility to eliminate potential OT bias is to generate similar mutations using two different sgRNAs (each with its own set of potential OT sites). This will confirm that the obtained phenotype is only due to the on-target event and not to OT mutation(s).

Several options are available to reduce the likelihood of generating OT mutations. They mainly rely on the many Cas protein variants that have been engineered to increase its specificity, or on the use of RNP complexes for quick transient expression of the system [22,50].

Generating transgenic hPSC clones is a stressful process for cells. Single-cell passaging, transfection, and clonal selection can favor the emergence and selection of aneuploid clones. The selection of clones with genomic abnormalities is also facilitated by the fact that several recurrent copy number variants (CNVs) can provide selective growth advantages to cells [51]. Therefore, the DNA integrity of the retained clones should be confirmed using standard cytogenetic procedures, G-banding, microarray, or next-generation sequencing [52]. If many clones have to be analyzed, recurrent CNV can be rapidly detected by qPCR or ddPCR to eliminate these clones before karyotyping [53].

Additional tests can be performed to assess the pluripotency status of the retained clones. The detection of pluripotency markers (e.g., octamer-binding transcription factor 4 [OCT4], NANOG, SRY-Box Transcription Factor 2 [SOX2], stage-specific embryonic antigen-4 [SSEA4]) by immunofluorescence analysis, qPCR, or cytometry analysis can rapidly confirm the pluripotency of the generated clones. As recent research has highlighted the potential preferential selection of p53-deficient hPSCs following CRISPR/Cas9 mutagenesis, it may be useful to check the p53 function or genomic sequence in the selected clones [42].

Finally, for each project, the functionality of the clones that have successfully passed all previous QC tests will have to be assessed. Tests will depend on the nature of the transgenic line that has been generated. It might require hPSC differentiation into the relevant cell type to obtain transgene expression. RNA and protein expression should be monitored in clones harboring indels and deletions. The expression of the modified allele or of the fluorescent reporters should be confirmed after KI.

## 4. Perspectives

Many protocols to improve the CRISPR mutagenesis rate have been published, but they often overlook the screening and validation steps required to characterize clonal lines. Here, we described a pipeline to efficiently design, generate, and characterize transgenic hPSC lines. Generating CRISPR/Cas9-edited hPSCs is a multi-step process that includes transfection, clone isolation, screening, and in-depth molecular validation. Each step can be performed using different protocols that are chosen in the function of their advantages and limitations. Careful experiment design and planning are crucial to select the most appropriate strategies to reduce the time and costs and to facilitate the generation of the desired hPSC lines.

The available techniques for in-depth characterization are limited by the high sequencing costs, but improving access to more comprehensive approaches, such as whole genome sequencing, should increase the reliability and safety of the edited cell lines in the future.

Furthermore, the development of new genome editing approaches, such as alternative CRISPR systems, base-editing [54], or prime-editing [55], will further facilitate hPSC engineering. It will also broaden the type of changes that can be introduced and expand the genome sequences that can be targeted. These systems should also reduce the risk of introducing unintended modifications. However, more research is needed to fully grasp the on- and off-target effects of these newer techniques in hPSC in order to adapt the screening strategies.

Human PSCs are valuable systems for basic research, disease modeling, pre-clinical, and now also clinical applications [56]. The ability to genetically manipulate these cells using CRISPR technologies to generate tailored transgenic lines that can be differentiated into any cell type is expanding the horizon of their possible applications. This will accelerate basic research by facilitating studies on genotype-phenotype correlations, and consequently increase the number and relevance of hPSC lines for disease modeling. Ultimately, these advances might be translated into autologous gene and cell therapies.

## Figures and Tables

**Figure 1 cells-09-01312-f001:**
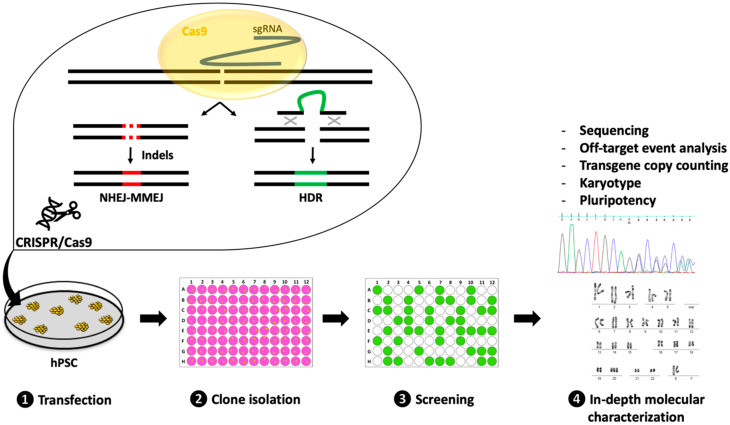
Pipeline to generate CRISPR-edited human pluripotent stem cell (hPSC) lines. Generating transgenic hPSC is a process that includes four mains phases: (**1**) Transfection of CRISPR reagents (single guide RNA, Cas9, and if required, a donor DNA template) in the parental hPSC line to introduce a targeted DNA double strand break (DSB). The DSB will be repaired by the endogenous DNA repair pathways. The non-homologous end-joining (NHEJ) and micro-homology-mediated end-joining (MMEJ) pathways can lead to the introduction of small insertions/deletions (indels), while the HDR pathway introduces exogenous nucleotides; (**2**) Transfected cells are isolated in separate wells to be expanded as clonal populations; (**3**) Following isolation, a high-throughput screening step is performed to select the correctly modified clones; (**4**) The selected clones are finally characterized using a combination of tests.

**Figure 2 cells-09-01312-f002:**
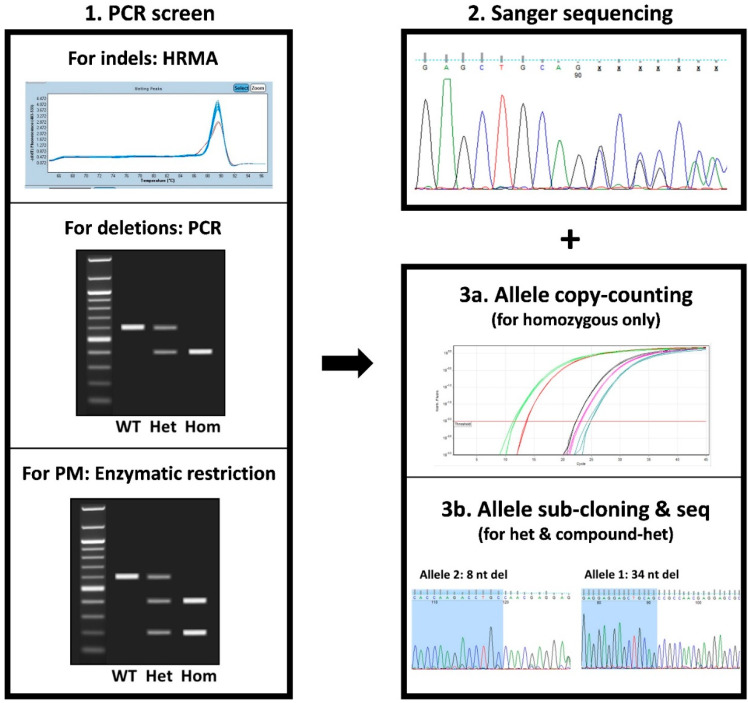
Screening strategies for indels, deletions, and small knock-in (KI). Screening to identify the CRISPR-edited hPSC clones harboring the desired indels, deletions, or small KI can be divided in three steps. The first step is a PCR-based screen to quickly detect clones with the on-target modification(s). For indels, the mutated clones can be identified by PCR followed by high resolution melt analysis (HRMA). To identify clones carrying a deletion, PCR with primers located at each side of the expected deleted sequence can be used. For clones carrying a small KI, if a restriction site has been added to the donor DNA template, enzymatic digestion of the PCR product that includes the targeted sequence will allow the detection of clones with on-target integration of the donor. Positive clones are then checked by Sanger sequencing to determine their exact sequence. This step is important to select clones with the desired modification(s) and to discard clones with unwanted events (i.e., in-frame events for KO projects, imperfect HDR events for a small KI project). Finally, in clones homozygous for the desired mutation, the transgene copy number is evaluated to ensure ploidy, whereas heterozygous and compound heterozygous clones are sub-cloned to determine the exact allelic sequences. PM, point mutation; WT, wild type; Het, heterozygous; Hom, homozygous; seq, sequencing; nt, nucleotides.

**Figure 3 cells-09-01312-f003:**
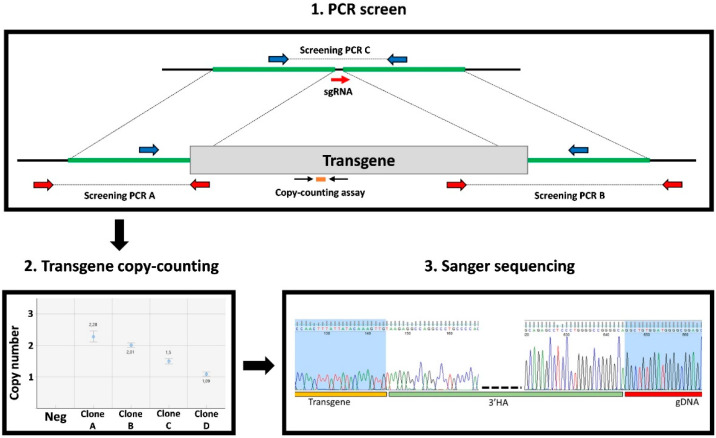
Screening strategies for large KI. Screening CRISPR-edited hPSC clones harboring large KI can be divided in three steps. The first step involves three PCR assays: (i) two PCR assays with primers that bridge between the transgene and the surrounding genomic DNA (screening PCR_A and B) to quickly detect clones with on-target integration of the transgene, and (ii) a third PCR assay using primers that surround the targeted locus (screening PCR_C) to detect alleles without transgene integration. Then, a transgene copy-counting assay by qPCR or ddPCR is performed to determine the transgene copy number. Combining the PCR and copy-counting results will allow classification of the clones in four categories: no transgene integration (PCR_A and B negative, PCR_C positive, copy-counting = 0), heterozygous integration (PCR_A and B positive, PCR_C positive, copy-counting = 1), homozygous integration (PCR_A and B positive, PCR_C negative, copy-counting = 2), other (all the other possibilities, including random insertion of the transgene and mosaic clones). For heterozygous and homozygous clones, the on-target integrated transgene sequence is verified by Sanger sequencing to ensure sequence integrity.

**Table 1 cells-09-01312-t001:** Advantages and limitations of the indel screening strategies.

Test (Reference)	Principle	Advantages	Limitations
T7E1 or Surveyor nuclease assay [35]	Enzymatic digestion of PCR heteroduplexes	Quick (few hours) Cheap Medium throughput Basic laboratory equipment	Sensitivity Medium throughput (needs also WT PCR product to detect homozygous clones) No information on the allele composition
HRMA [43]	Melting curve variation	Quick (few hours) Cheap High throughput Sensitive Basic laboratory equipment	Short amplicons required, might miss larger deletion events No information on the allele composition
IDAA [44]	Indel detection by amplicon analysis	Quick (few hours) High throughput Sensitive Information on the allele composition	More expensive Specific laboratory equipment
ddPCR [45]	Indel detection by drop-out of a labeled probe	Quick (few hours) Medium throughput Sensitive	More expensive Specific laboratory equipment No information on the allele composition

Several strategies are available to identify hPSC clones harboring the desired indel(s). WT: wild-type.

**Table 2 cells-09-01312-t002:** In-depth molecular characterization tests.

Test	Project Type	Purpose	Methods
DNA integrity	All	Identification of unwanted genomic abnormalities/rearrangements	Screening: qPCR or ddPCR Final characterization: G-banding/microarray/NGS
OT mutation	All	Identification of potential mutagenesis events at off-target sites	HRMA or PCR of the top 5-10 OT sites/HRMA or PCR of OT with up to 3 MMs/whole exome/whole genome
Pluripotency	All	Confirming the pluripotency state of the transgenic iPSC lines	IF/cytometry/embryoid bodies/teratoma
Plasmid integration	All projects in which plasmids are used	Confirming the absence of plasmid backbone integration in the clone’s genome	PCR/qPCR/ddPCR
ssDNA donor integration	All projects in which ssDNA are used	Identification of unintended integration of the ssDNA donor oligo(s)	qPCR/ddPCR
p53 function	All	Confirming p53 function	Sequencing

Following screening, clones need to be further characterized to ensure their integrity. NGS: next generation sequencing; MMs: mismatches; IF: immunofluorescence.

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
