# Peer review of "Pipeline for the Generation and Characterization of Transgenic Human Pluripotent Stem Cells Using the CRISPR/Cas9 Technology"

_cells, 2020, doi:10.3390/cells9051312_

Round 1

Reviewer 1 Report

In the present review Miannè and colleagues provide an interesting overview of the main available strategies to generate transgenic human pluripotent stem cells (hPSCs). In particular they described a pipeline to efficiently design, generate and characterize CRISPR/Cas9-edited hPSC lines in function of the introduced mutation.

The paper is well-written and the authors highlight the proposed issues in a comprehensive way. Nevertheless, I suggest to argue in a short paragraph the challenges and future perspectives regarding the use of  CRISPR/Cas9 as an important technology in stem cell research field.

Author Response

We would like to thanks the Reviewer for the positive feedback. Regarding the suggestion, we have replaced the “Conclusion” section by a “Perspectives” section and have added the following text:

“The available techniques for in-depth characterization are limited by the high sequencing costs, but improving access to more comprehensive approaches, such as whole genome sequencing, should increase the reliability and safety of the edited cell lines in the future.

Furthermore, the development of new genome editing approaches, such as alternative CRISPR systems, base-editing [54] or prime-editing [55], will further facilitate hPSC engineering. It will also broaden the type of changes that can be introduced and expand the genome sequences that can be targeted. These systems should also reduce the risk of introducing unintended modifications. However, more research is needed to fully grasp the on- and off-target effects of these newer techniques in hPSC in order to adapt the screening strategies.

Human PSC are valuable systems for basic research, disease modeling, pre-clinical and now also clinical applications [56]. The ability to genetically manipulate these cells using CRISPR technologies to generate tailored transgenic lines that can be differentiated in any cell type is expanding the horizon of their possible applications. This will accelerate basic research by facilitating studies on genotype-phenotype correlations, and consequently increase the number and relevance of hPSC lines for disease modeling. Ultimately, these advances might be translated into autologous gene and cell therapies.”

Reviewer 2 Report

In this manuscript, the authors summarized the main strategies and proposed an efficient pipeline to generate, screen, and characterize CRISPR/Cas9-edited hPSC lines obtained by knock-out (KO), small or large knock-in (KI), and deletions of genetic elements. This manuscript provides an overview of the available strategies to generate transgenic hPSC lines with optimal quality assessment and described a pipeline to efficiently design, generate, and characterize transgenic hPSC lines. There are several errors throughout the manuscript. For example, check the errors in lines 237-239 (page 6 of 15). The authors are strongly required to check and edit any errors throughout the manuscript. More importantly, authors need to provide more detailed and descriptive information (summary) on the various recently developed techniques for the efficient generation, screening, differentiation, and characterization of genome-edited hPSCs. For example, a more detailed description of the techniques used in references [12] and [13] in lines 234-239 (page 6 of 15).

Author Response

We would like the thanks the Reviewer for the feedback. For the lines 234-239, we have amended the text to correct the error and provide more details regarding ref [12] and [13]. The following section was removed:

“For some projects, clone isolation can be facilitated by adding a positive selection marker to specifically select the transfected, or correctly engineered cells [12]. For instance, a fluorescent reporter to specifically isolate transfected cells by fluorescence-activated cell sorting can be added. Alternatively, an antibiotic resistance cassette can be appended to eliminate non-transfected cells before cloning. For KI projects leading to expression of the transgene in hPSC, then addition of a positive selection cassette will help selecting cells with the desired KI event [13].”

And replaced by:

“For some projects, clone isolation can be facilitated by adding a positive selection marker (fluorescent reporter, or antibiotic resistance gene) to specifically select the transfected, or correctly engineered cells.[12]. Cells that express a fluorescent reporter can be selected by fluorescence-activated cell sorting (FACS), whereas cells that include an antibiotic resistance gene will survive antibiotic selection. These positive markers are usually transiently expressed as part of the transgene and removed at a later stage, or are delivered with the transient Cas9 expression system. If the project aim is to introduce a tagged protein or a gene the expression of which is controlled by an endogenous promoter, the strategy can be adapted depending on whether this gene is expressed or not in hPSC. Indeed, if the protein with a fluorescent tag is expressed in hPSC, correctly targeted cells can be detected and isolated by FACS [13]. Conversely, FACS-based screening is not possible if the tagged protein is not expressed in hPSC. Innovative approaches have been proposed to overcome this issue. For example, Roberts et al. described a strategy based on a transgene with a constitutively expressed mCherry fluorescence selection cassette that allows selection by FACS of clones that carry the transgene. This was followed by excision of the selection cassette using CRISPR/Cas9 and MMEJ repair [12].”

Additionally, errors were checked and corrected throughout the text.  

Round 2

Reviewer 2 Report

Authors tried to check and edit the errors throughout the manuscript. Authors also tried to provide more information on the various recently developed techniques for the efficient generation, screening, differentiation, and characterization of genome-edited hPSCs.